# *In Silico* identification of angiotensin-converting enzyme inhibitory peptides from MRJP1

Rana Adnan Tahir[1,2], Afsheen Bashir[3], Muhammad Noaman Yousaf[3], Azka Ahmed[2], Yasmine Dali[4], Sanaullah Khan[5], Sheikh Arslan Sehgal [6]*

**1** Key Laboratory of Molecular Medicine and Biotherapy in the Ministry of Industry and Information Technology, Department of Biology, School of Life Sciences, Beijing Institute of Technology, Beijing, China, **2** Department of Biosciences, COMSATS University Islamabad Sahiwal Campus, Sahiwal, Pakistan, **3** Khyber Girls Medical College, Hayatabad, Peshawar, Pakistan, **4** State Key Laboratory of Membrane Biology, Institute of Zoology, Chinese Academy of Sciences; Beijing, China, **5** Department of Zoology, University of Peshawar, Peshawar, Pakistan, **6** Department of Bioinformatics and Biotechnology, Government College University Faisalabad, Faisalabad, Pakistan

* arslansehgal@yahoo.com

**Data Availability Statement:** All relevant data are within the manuscript and its Supporting Information.

## Abstract

Hypertension is considered as one of the most common diseases that affect human beings (both male and female) due to its high prevalence and also extending widely to both industrialize and developing countries. Angiotensin-converting enzyme (ACE) has a significant role in the regulation of blood pressure and ACE inhibition with inhibitory peptides is considered as a major target to prevent hypertension. In the current study, a blood pressure regulating honey protein (MRJP1) was examined to identify the ACE inhibitory peptides. The 3D structure of MRJP1 was predicted by utilizing the threading approach and further optimized by performing molecular dynamics simulation for 30 nanoseconds (ns) to improve the quality factor up to 92.43%. Root mean square deviation and root mean square fluctuations were calculated to evaluate the structural features and observed the fluctuations in the timescale of 30 ns. AHTpin server based on scoring vector machine of regression models, proteolysis and structural characterization approaches were implemented to identify the potential inhibitory peptides. The anti-hypertensive peptides were scrutinized based on the QSAR models of anti-hypertensive activity and the molecular docking analyses were performed to explore the binding affinities and potential interacting residues. The peptide "EALPHVPIFDR" showed the strong binding affinity and higher anti-hypertensive activity along with the global energy of -58.29 and docking score of 9590. The aromatic amino acids especially Tyr was observed as the key residue to design the dietary peptides and drugs like ACE inhibitors.

## Introduction

Pulmonary arterial hypertension (PAH) affects the small pulmonary arterioles, which lead to a progressive disease of the lung vascular system. The progressive narrowing of the blood vessels is a collective effect of increased contractility of the small pulmonary arteries, remodeling, and proliferation of endothelial smooth muscle cells and endothelial dysfunction [1].

**Funding:** The authors received no specific funding for this work.

**Competing interests:** The authors have declared that no competing interests exist.

PAH is transmitted through an autosomal dominant trait with mitigated trenchancy. The mutations in the bone morphogenetic protein receptor type-II (BMPR2) elucidate 70% of the hereditary cases while 20% of the cases have unknown reasons [2, 3]. BMPR2 belongs to the super-family of TGFb/BMP [4] and its heterozygous alterations occur in the transmissible PAH [5, 6] leads to the illness [7]. The hereditary PAH is localized at chromosome 2q33 [8, 9]. The nonsense, frameshift and missense mutations in BMPR2 lead to change the bone morphogenetic protein and TGF-b1/SMAD signaling pathways, which ultimately cause escalation instead of apoptosis of the vascular cells [6, 7, 10–12]. The system modifications entailed in the cardiovascular attunement are possibly associated with the commencement and conservation of the blood pressure elevation [13].

Angiotensin-converting enzyme (ACE) is a vital constituent of the renin-angiotensin system (RAS), arbitrating various systemic and local effects in the cardiovascular system. The ACE synthesis in somatic tissues endothelium as a transmembrane protein comprising of two active domains which are inhibited by ACE inhibitors [14]. ACE peptides as inhibitors are extensively studied in different bioactive peptides [15–18] for therapeutic purposes. The conversion of ACE transmutes angiotensin I to angiotensin II is a dynamic vasoconstrictor and a vital enzyme in the modulation of blood pressure and body fluids. It is also involved in the anatomization of bradykinin to dilate the blood vessels [19].

The ACE function could induce the vasoconstriction and progression of hypertension and related pathological manifestations. ACE suppression is considered as an essential approach in regulating hypertension [20]. The synthetic or celluloid ACE inhibitor drugs have side effects including a dry cough, skin rashes or erythema, taste turbulences and the modifications in serum lipid metabolism [21]. The commercially available ACE inhibitor drugs are discouraged and food protein-derived ACE inhibitory peptides are preferred [18, 22, 23] for effective therapies. The amino acid residues determine the inhibitory potency of ACE inhibitory peptides such as the existence of hydrophobic and positively charged amino acids [24, 25]. The purpose to assess the food proteins from primary food products as precursors in producing ACE inhibitory peptides facilitates to develop a principle for proper selection of substrate protein. The high-potential food and the sedentary lifestyle are known to trigger hypertension [26].

The current work demonstrates the *in silico* identification of potential anti-hypertensive peptides from honey protein MRJP1. Computational approaches have shown considerable success in research methodologies to solve biological problems [27]. After the successful identification of computational drugs and drug targets in neurological disorders [27–32] and cancer [33–36], researchers also utilized the computational approaches to design epitope-based peptide vaccines through immunoinformatic approaches [37]. The 3D model was built by using homology modeling and threading based approaches followed by the Molecular Dynamic (MD) simulations to optimize and analyze the structural features of a model for protein-peptide docking analyses. The screening for ACE inhibitory peptides was performed to identify the potential anti-hypertensive peptides. The observed anti-hypertensive peptide-protein interactions may serve to replace the drugs by dietary peptides and to narrow down the diverse combinatorial search space.

## Results and discussion

The objective of the current research was to identify the potential anti-hypertensive peptides derived from MRJP1. The retrieved sequence of MRJP1 was used to identify the appropriate templates but the query coverage and sequence identity against suitable templates were not satisfactory to build a model through a comparative modeling approach. The top-ranked template belongs to Salivary protein having only 25% identity and 61% query coverage was

observed, therefore the threading based approach was utilized through I-Tasser for structure prediction of MRJP1.

The top-ranked five models were predicted by using the templates with higher similarity identified through the threading alignments. It was observed that the template protein (PDB ID: 3q6k) has a resolution of 2.52 Å structure (Salivary protein) and showed the highest confidence score of 0.54. The salivary protein belongs to the MRJP protein family [38] and the first structurally characterized member of the family that is being utilized in MRJP1 structure prediction. The homologous templates for evolutionarily related proteins are identified through the sequence profile analyses [39] and considered as reliable for the prediction of high-resolution structures. The non-homologous proteins may also have the similar structures, and threading approaches [40, 41] have ability to match the query sequences onto the available structures with the aim of identifying the similar folds to the query even though there is no evolutionary relationship among the template protein and the query sequence. The models predicted through homology modeling and threading approaches with the RMSD range of 2–5 Å from distant templates that can be utilized for functional analyses and the identification of the active site residues [42–45]. MD simulation has been utilized for the *ab initio* structure prediction [46] to simulate the folding of the protein, while the template-based structure prediction is considered as one of the most reliable approaches [47–54].

Numerous models were predicted by utilizing a homology modeling and threading based approach and all the predicted models were evaluated critically. The model showed 78.53% quality factor and further subjected for MD simulations to optimize and extract the structural fluctuations throughout the 30 ns and it was observed that the quality factor was improved up to 92.43% (S1 Fig) and while 98.3% residues appeared in favorable regions.

## 2.1 Molecular dynamics simulation

The predicted structure of MRJP1 was subjected to MD simulations applying ensemble, temperature and appropriate solvent molecules. The constant temperature for 300K, 1atm pressure and heating for 500 ps were applied for simulation experiments in initial equilibration. The steric energy constraints were eliminated or reduced through energy minimization. Newtonian's dynamics equilibrated the system to locate a thermally bound state, which leads to the production runs and simulations also deliver ensembles of structure to analyze the results. The conformational changes in the MRJP1 structure have been concluded from macroscopic features. The conformational variations of the MRJP1 structure were analogously determined at 0 ns, 10 ns, 20 ns, and 30 ns. Three major physical properties comprising RMSD, RMSF, and B-factor of the simulated system were calculated to analyze the conformational changes in the hydrated environment.

**2.1.1 Root mean square deviation (RMSD).** The atomic position of RMSD was calculated by considering the predicted structure of MRJP1 as a foremost model to find out the sustainability and convergence of the MD simulations. The 30 ns runs of molecular dynamics denoted that the RMSD of Cα-atoms as a function of simulation time (Fig 1). The results indicated that the RMSD values showed minimal fluctuation throughout the simulations studies. The high variation of atoms along with the residues close to NTD and CTD were observed. Overall, the stability of the structure was observed in 30 ns particularly at the end of the simulation, thus the simulated model was utilized for further processes. RMSD analyses of MRJP1 have shown no major fluctuations throughout the 30 ns simulations. The protein showed some higher fluctuations only at the start of the simulation while the stability of the structure was observed at the end of the simulation system.

**2.1.2 Root mean square fluctuation (RMSF).** The RMSF analysis of a protein about their conformations is a significant mark of many biological processes which includes complex

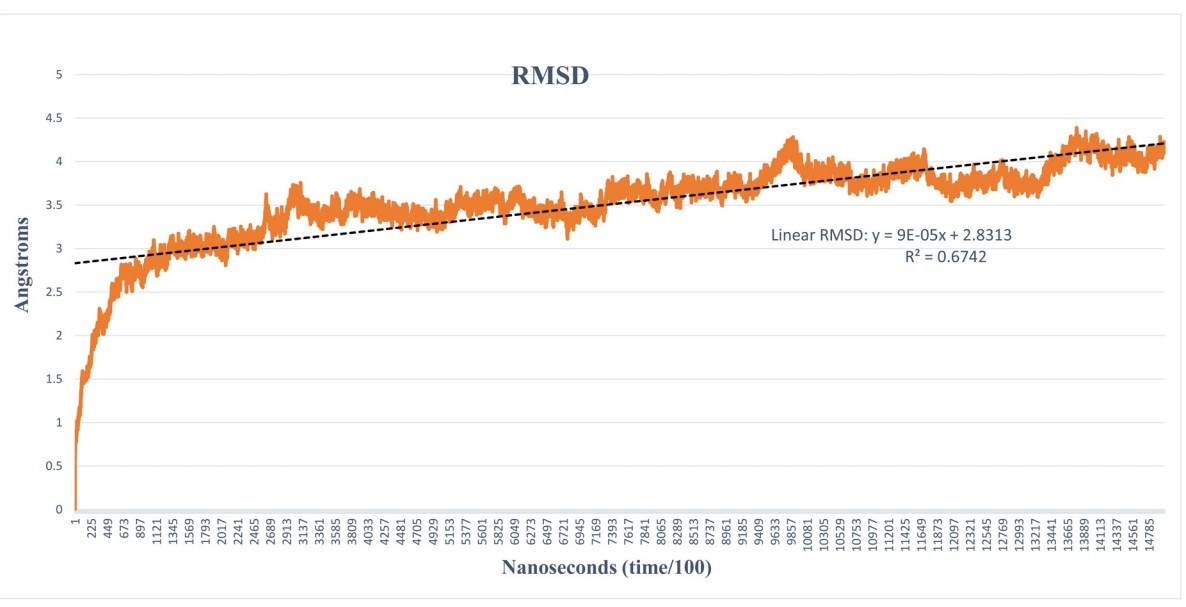

**Fig 1. Root mean square deviation graph vs. time; the graph showed the minimal fluctuations throughout the simulation runs and structural stability and optimization were achieved with respect to time.**

recognition, protein activity and macromolecular recognition [55]. The RMSF graph was computed for each residue of Cα-atoms, while the overall MRJP1 structure exhibited an advanced fluctuation level. The RMSF graph demonstrated the residual fluctuations of the MRJP1 model over 30 ns timeframe (Fig 2) and four major fluctuation peaks were observed. The first major residual fluctuation was observed from 30–67 (37 residues) amino acids, while second, third and fourth were 134–153 (19 residues), 228–256 (28 residues), and 374–405 (29 residues) amino acids respectively.

**2.1.3 B-Factor.** The applications of computational advances are to anticipate the thermal motion that examines the obscure structure of the proteins with dynamic attributes. The polypeptide backbones and side chains of MRJP1 structure were persistent in motion owing to kinetic energy and thermal motion of atoms. The fluctuations of the atoms regarding their average positioning were reflected by B-factors of protein structure and provided significant evidence about the protein dynamics. RMSD and PMSF plots indicated the stability of the model and only a few structural fluctuations were observed at residues level. It has also been verified through secondary structure analysis that there were few coils (irregular) elements along helices and sheets. Moreover, the observed B-factor analyses were in favor of higher values at corresponding positions anticipating that the MRJP1 structure is reliable for further analyses (Fig 3).

## 2.2 Structural analyses

The structural analyses were performed of simulated MRJP1 model at varying degrees of MD simulations such as 0 ns, 10 ns, 20 ns, and 30 ns. The structural fluctuations along with

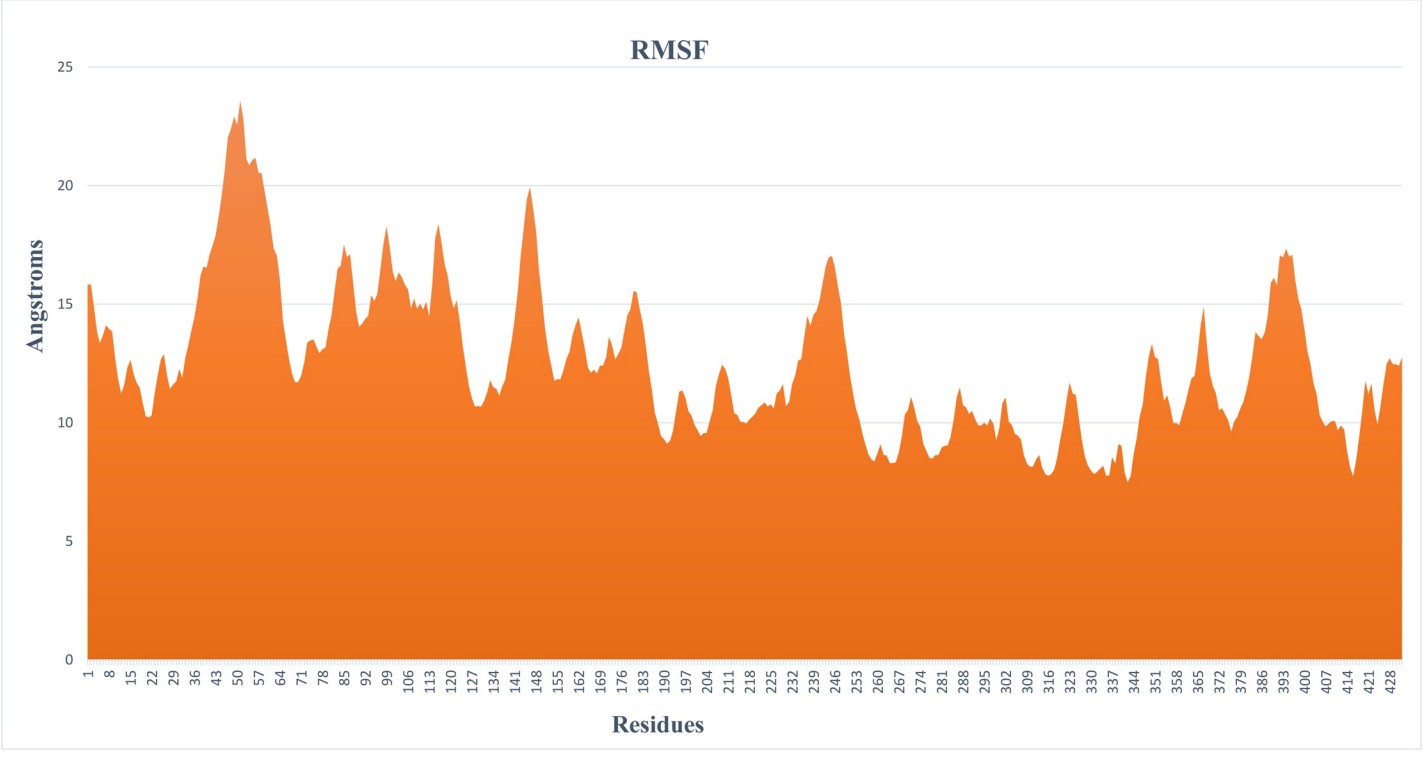

**Fig 2. RMSF fluctuation graph showed the variations of individual residues from 0–30 ns.**

differences in the number of helices and sheets were observed in structural analyses (Fig 4). The most prominent difference in terms of improvement was the quality factor and the structural stability from 0 ns, 10 ns, 20 ns, and 30 ns structures as 78.53%, 85.37%, 89.14%, and 92.43% respectively. The number of alpha-helices at 0 ns and 30 ns were same as ten (10) helices but vary in residues length from 33 to 39 residues respectively, while the structure at 10 ns and 20 ns contains 14 and 11 helices respectively. On the other hand, 23 beta-sheets were observed at 20 ns and 30 ns while the structure at 0 ns and 10 ns comprise 21 and 24 beta-sheets respectively. The fluctuations in the number and lengths of the secondary structural elements were observed in the simulated model that greatly influenced the structural quality. The terminal directions of the structure have changed during the simulation analyses. The N and C terminals in the unrefined structure were embedded in the structure and projected inwards. The refined structure has terminals projected out of the protein structure with clear ends. It was also seen that the pattern similarity in overall structure and protein model stability incremented with MD simulations.

## 2.3 Derived peptides

The peptides were manually derived based on the properties of the interacting residues and structural characterization of the amino acids; the peptides considered for the current study were specifically including di-peptides. The criteria for the selection of di-peptides includes both of the amino acids either belong to a hydrophobic group or bulky hydrophobic. The peptides were derived by using the peptide cutter with two enzymes pepsin and trypsin individually, structural characterization and AHTpin server based on support vector machine score (SVM). The peptides derived from the applied techniques are mentioned in Table 1 with the cleavage site, peptide length, and SVM score.

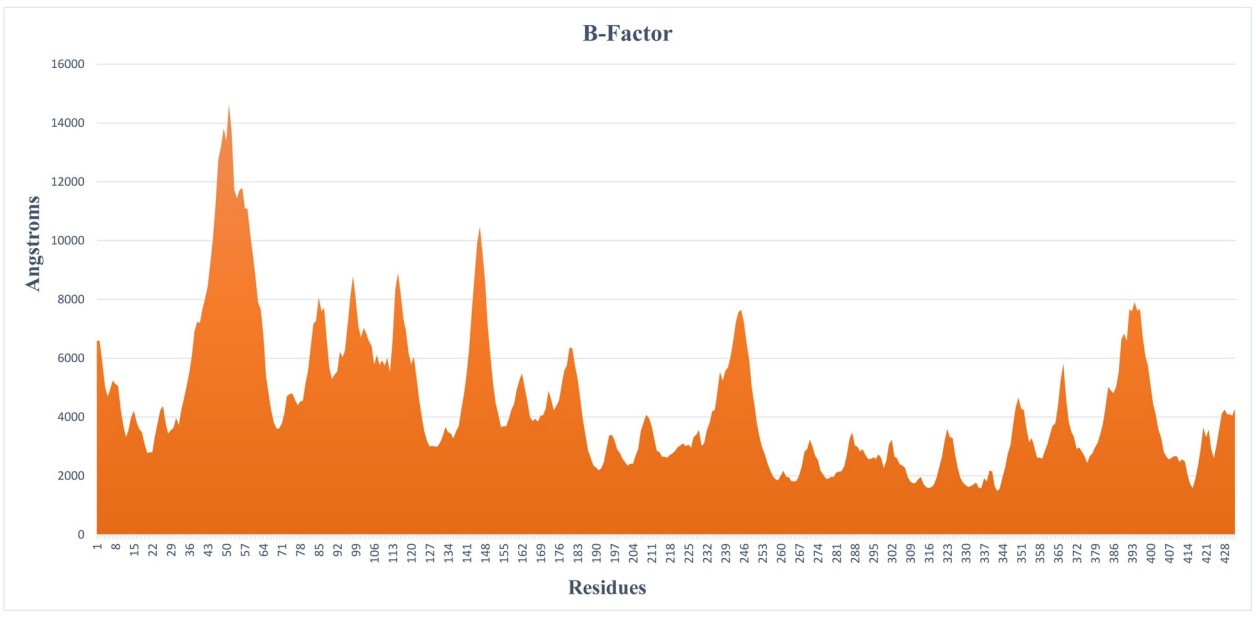

**Fig 3. The B-factor analysis represents the fluctuations of the atoms regarding their average positioning.**

All the derived peptides were evaluated by the regression models of SVM score and the leading peptides were docked with ACE to identify the high binding affinities. SVM regression model was built for di- and tri-peptides, while SVM classification models for peptides have more than three residues. The applied methods were based on the nature of amino acids, atomic composition and chemical descriptors (15,537) while trained by the machine learning techniques to evaluate through regression and classification methods.

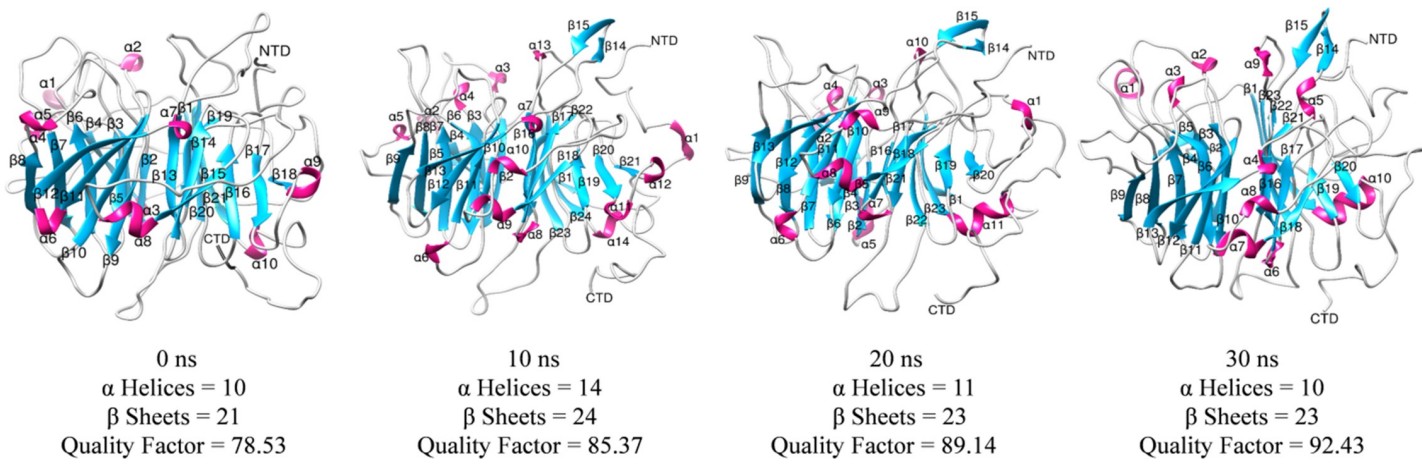

**Fig 4. Structural details of the simulated structure of MRJP1 at 0 ns, 10 ns, 20 ns, and 30 ns.**

**Table 1. Derived peptides having anti-hypertensive activity.**

| Technique | | Cleavage Site | Peptide | Peptide Length | SVM Score | Anti-hypertensive Peptide inhibitor | | | |
|---|---|---|---|---|---|---|---|---|---|
| | | | | | | Cleavage Site | Peptide | Peptide Length | SVM Score |
| Proteolysis | Pepsin | 110 | LLQPYPDW | 8 | 1.47 | 265 | LYYSPVASTSLYY | 13 | 1.74 |
| | | 294 | QQNDIH | 6 | 0.35 | 264 | NLYYSPVASTSLY | 13 | 1.66 |
| | | 135 | AIDKCDRL | 8 | 0.04 | 266 | YYSPVASTSLYYV | 13 | 1.52 |
| | | 369 | PHVPIF | 6 | 1.72 | 262 | TNNLYYSPVASTS | 13 | 1.49 |
| | | 237 | YDPKF | 5 | 0.76 | 102 | PLLQPYPDWSFAK | 13 | 1.48 |
| | Trypsin | 114 | VGDGGPLLQPYPDWSFAK | 18 | 0.39 | 101 | GPLLQPYPDWSFA | 13 | 1.41 |
| | | 62 | QDAILSGEYDYK | 12 | 0.65 | 151 | SPKLLTFDLTTSQ | 13 | 1.38 |
| | | 166 | LLTFDLTTSQLLK | 13 | 0.30 | 268 | SPVASTSLYYVNT | 13 | 1.38 |
| | | 371 | EALPHVPIFDR | 11 | 1.52 | 100 | GGPLLQPYPDWSF | 13 | 1.36 |
| Structural Characterization | | 175 | AV | 2 | 3.10 (pIC$_{50}$) | 96 | KVGDGGPLLQPYP | 13 | 1.36 |
| | | 53 | AI | 2 | 5.47 (pIC$_{50}$) | 263 | NNLYYSPVASTSL | 13 | 1.33 |
| | | 212 | GL | 2 | 2.60 (pIC$_{50}$) | 267 | YSPVASTSLYYVN | 13 | 1.30 |
| | | 255 | GM | 2 | 2.85 (pIC$_{50}$) | 97 | VGDGGPLLQPYPDWS | 15 | 1.20 |
| | | 85 | GV | 2 | 2.34 (pIC$_{50}$) | 99 | DGGPLLQPYPDWS | 13 | 1.16 |
| | | 137 | VL | 2 | 4.89 (pIC$_{50}$) | 258 | LSPMTNNLYYSPV | 13 | 1.11 |
| | | | | | | 257 | ALSPMTNNLYYSP | 13 | 1.05 |
| | | | | | | 164 | LLQPYPDWSFAKY | 13 | 1.03 |

Regression analyses were conducted to correlate the chemical descriptors and biological activity (pIC50) of small peptides for the pIC50 prediction of novel peptides.

Di-peptides and tri-peptides belong to a small class of peptides but separate regression models were implemented for each method to predict the biological activity. The classification models predicted the special type of peptides either AHT or non-AHT based on the descriptors of the training set. Mainly, PubChem, CDK-fingerprint, XLogP, electrotopological state atom type, and auto-correlation descriptors were implemented to develop the di-peptide QSAR model while tri-peptide QSAR model was developed primarily by KlekotaRoth fingerprint count, PubChem fingerprint, CDK graph only fingerprint and extended fingerprint descriptors [56]. The biological activity of di-peptides including AV, AI, GL, GM, GV & VL from MRJP1 was validated and evaluated by AHTpin. The reliability and bioactivity of all the derived peptides from MRJP1 were validated by AHTpin.

The lead anti-hypertensive peptides were selected on the basis of SVM scores for molecular docking analyses. The protein-peptide molecular docking analyses were performed and ACE was utilized as receptor against all the derived peptides to determine the binding position and orientation (S1 File). The docking analyses were performed by using the segmentation technique to identify and scrutinize the patches to evaluate the binding conformations and give a score to geometric complementary shapes. The docking complexes were ranked by the observed docking score and top ten ranked peptides having highest binding affinities were selected (Table 2) for further binding interactional studies through PyMol and UCSF Chimera (Fig 5). It was observed that the peptide "EALPHVPIFDR" from all the scrutinized peptides showed docking score of 9590 and effective binding affinity. The interesting fact was observed

**Table 2. Protein-peptide interactions along with docking scores and binding residues.**

| Peptide | SVM Score | PatchDock Score | Global Energy (kcal/mol) | ACE Binding Residues |
|---|---|---|---|---|
| EALPHVPIFDR | 1.52 | 9590 | -58.29 | Tyr62, Ala63, Asn66, Asn70, Ile88, Lys118, Glu123, Met223, Val351, His353, Ala354, Ser355, Ala356, Trp357, Asp358, Tyr360, Lys368, Glu384, Phe391 Arg402, Glu403, Phe512, His513, Ser516, Ser517, Val518, Tyr520, Arg522, Tyr523, Zn701 |
| NLYYSPVASTSLY | 1.66 | 11060 | -52.04 | Trp59, Tyr62, Asn66, Asn70, Leu81, Lys118, Val119, Gln120, Asp121, Glu123, Arg124, Leu139, Leu140, Tyr213, Met223, Val351, His353, Ser355, Trp357, Lys368, Arg402, Glu403, Phe512, Ser516, Ser517, Val518, Pro519, Phe570 |
| PHVPIF | 1.72 | 6968 | -47.47 | Trp59, Tyr62, Ile88, Thr92, Lys118, Glu123, Arg124, Tyr360, Arg402, Glu403, Pro519, Arg522 |
| LYYSPVASTSLYY | 1.74 | 10388 | -33.24 | Trp59, Tyr62, Asn66, Asn70, Lys118, Asp121, Glu123, Arg124, Ser219, Trp220, Ser222, Tyr213, Met223, Ser355, Ala356, Trp357, Tyr360, Glu403, Asn406, Pro407, Ser516, Ser517, Val518, Pro519, Arg522, Phe570, Zn701 |

that the scrutinized top-ranked peptide was embedded in the receptor surface and engaged the binding domain. The anti-hypertensive peptides EALPHVPIFDR, NLYYSPVASTSLY, PHVPIF, and LYYSPVASTSLYY showed least binding energies may have the potential to behave as ACE inhibitors. The binding interactions of the selected peptides revealed that the Tyr residue is the most common interacting residue that behaved as an ACE inhibitor and has the potential to be a potent drug target.

Majority of the therapeutic agents attain their outcomes by binding and modify the functions of the target proteins. Traditionally, the binding within small cavities and catalytic sites inhibition exhibit the high affinity and successful therapeutics by compounds [57].

Food is considered as a source of nutrients and energy essentials to sustain the appropriate functions of the body. Now, scientists are trying to identify the novel characteristics of food constituents that may assist to overcome the numerous 'diseases of civilization'. The mutual objectives of nutritionists, food manufacturers and researchers are to focus the proteins that have the origins of ACE inhibitors, to enhance their bioactivity and formulating those as commercial food to improve the human health [58]. In recent years, peptides have gained demanding attention in the pharmaceutical research for being highly efficacious, selective and relatively safe. More than a few hundreds of novel peptide therapeutics are currently being evaluated in pre-clinical and clinical trials while over 60 peptides have reached the market for different therapies [59].

Various side effects such as cough, headache, dizziness, and angioedema of synthetic anti-hypertensive drugs have been reported [60]. Therefore, the identification of potential anti-hypertensive biopeptides from foods gained attention [61]. The peptides of anti-hypertension have been reported in various dietary sources including egg, milk, meat, potato, wheat, soya beans, and vegetables. The synthetic compounds also occur as ACE inhibitors for hypertension therapies, although synthetic drugs contain adverse effects. So, the inclination towards nature-derived anti-hypertensive molecules is highly desired. The *in silico* identification of ACE inhibitory peptides from honey protein was performed which is considered as a source of anti-hypertension in the form of the ACE inhibitor [17].

The functional and nutritional features of dietary proteins have been studied over decades. The physiological consumption of amino acids after digestion and protein composition exhibit the nutritional characteristics [62]. Glycine was found as a predominant amino acid in AHTs server analysis and possesses two residues, revealed through amino acid composition investigation [56].

The proteolytic processing of food proteins leads to the production of active and bioactive peptides that performs various physiological functions of the body. These bioactive peptides

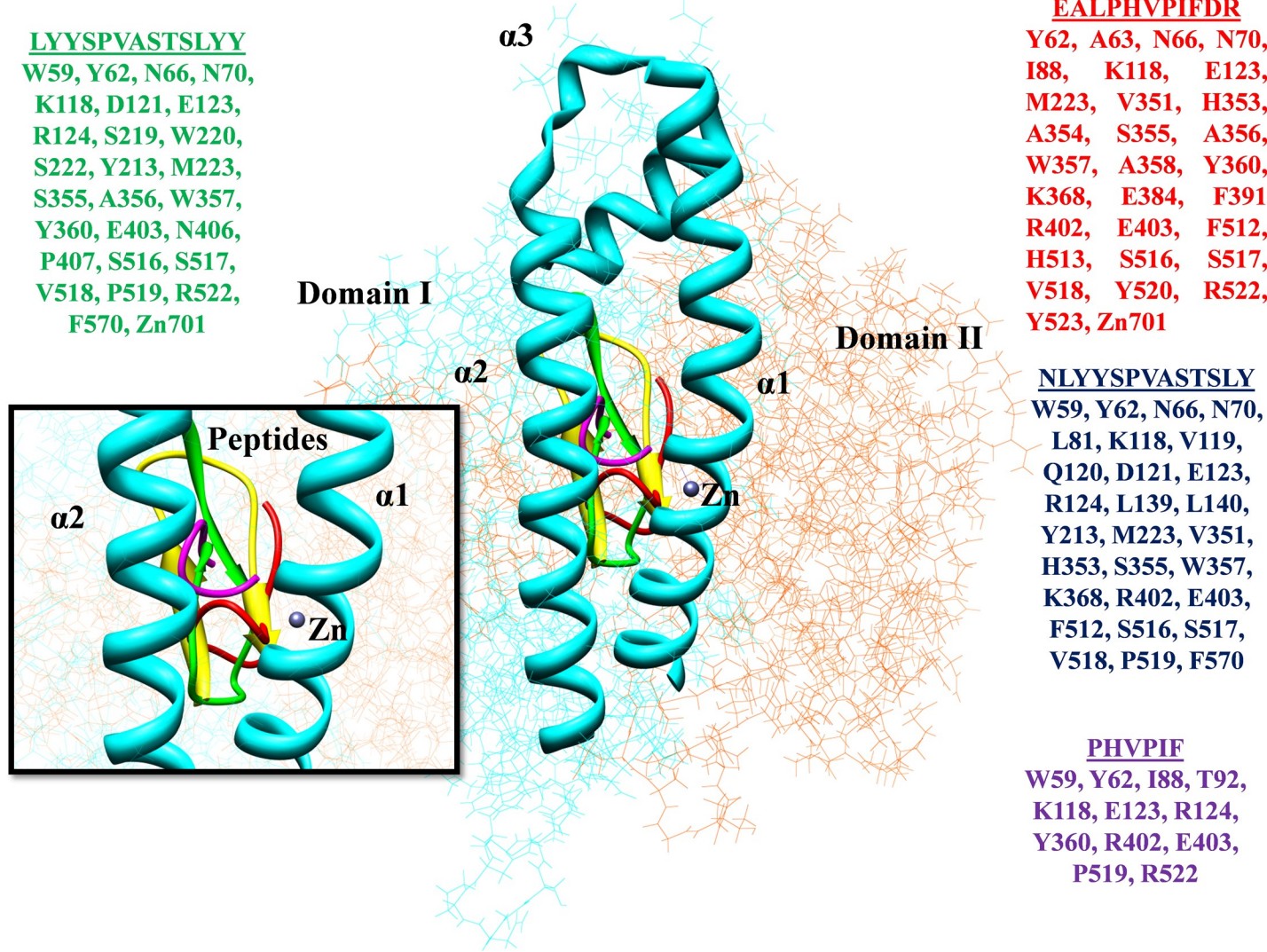

**Fig 5. Interacting residues of the ACE-peptides are represented in different colors.** The crystal structure of human ACE (PDB ID: 1O8A) protein is divided into two domains as Domain I (N-terminal) (37–291 amino acids) represented in cyan color while Domain II as C-terminal domain is presented in orange color (292–625 amino acids). The N-terminal lid appeared as the α1, α2, and α3 exhibiting the active site of protein along with the Zn binding site. The scrutinized peptides showed the interactions at binding sites and represented in different colors along with interacting residues.

may act as an opioid antagonist, agonists, anti-hypertensive agents, and moreover anti-cancer, anti-thrombotic, anti-microbial, immune-modulating and anti-oxidative activity have been reported. The bioactive peptides may be utilized in functional food components due to their therapeutic potentials [63].

The peptides are preferred over the small compounds due to their structural compatibility, small size and ability to interrupt protein-protein interfaces. The rational methods have a key hindrance to design effective peptide ligands for the development of potential drugs. However, numerous computational techniques have evidenced the structural and functional insights into the architecture of protein-peptide interfaces for the rational peptide design approach. These methods help to fulfill the vision of computationally designed peptides for therapies through the high-resolution structures of protein-peptide complexes [64–66].

*In vivo* studies have found that anti-hypertensive effects can be attained in humans through peptides especially di- and/or tri-peptides [67NR, 68]. Hata *et al.*, [68] demonstrated the effectiveness of Ile-Pro-Pro and Val-Pro-Pro on blood pressure (BP) regulation. They hypothesized that stimulation in aortas along with circulatory ACE inhibition would be the reason for that effect [69]. It has also been reported that the intake of bioactive Val-Tyr di-peptide led to a significant reduction of systolic BP after 1 week on mildly hypertensive subjects [67]. These discoveries strongly recommended that the renin-angiotensin system suppression by bioactive smaller peptides play a significant role in the regulation of BP.

The ACE inhibition was greatly enhanced through gastrointestinal protease hydrolysis of royal jelly by trypsin followed by pepsin and chymotrypsin [70]. These analyses reported that the inactive royal jelly proteins might be an effective ACE inhibitor to regulate the BP and new peptide inhibitors in gut formed through gastrointestinal proteases would be more significant. Uno *et al.* [71] documented that consumption of royal jelly hydrolysate by trypsin and pepsin amplified the hemoglobin levels and reduced the higher cholesterol levels in human beings. Therefore, the royal jelly is considered as a beneficiary to improve the homeostasis.

Ohashi *et al.* [72], derived peptides from the royal jelly glycoproteins and demonstrated that most of the isolated peptides have aromatic amino acid residues as Phe and Tyr at C-terminus exhibited the strong inhibitory activity. Cheung *et al.*, [73] confirmed the inhibition potential of these aromatic peptides in their research and observed the additional ACE inhibition for peptides with Trp-Tyr-Phe at the C-terminus. It has also been reported that the peptides having Ile-Val-Tyr residues extracted from the royal jelly hydrolysate with the highest ACE inhibitory contribution rate of 16.9% in addition to wheat germ hydrolysate [74].

Okunishi *et al.* [75] elaborated the long-term oral therapeutic drug, spirapril that suppresses the ACE activities in blood vessels and induce the extended depressor effects. Their analyses showed that few of the natural inhibitory peptides, specifically royal jelly peptides could gather at the vessels and exert a regulation of secretion for active elements including prostaglandins or endothelin and nitric oxide [76]. The royal jelly protein has the ability to produce plenty of ACE inhibitory peptides throughout the digestion to reduce the depressor effect and it was a latent natural source along with *vivo* anti-hypertensive effects.

The development of peptide-based therapeutics is of great interest and has rapid growth [77–79]. Currently, a robust approach has been evolved that incorporates topographical, conformational, dynamic and structural considerations to design the peptides for drugs, drug molecules, and biological tools. Current developments to understand the chemistry of life, specifically molecular biophysics, proteomics, genomics, and molecular biology have described that the macromolecular-peptide interactions establish the key physiochemical processes whereby living mechanisms are modulated and controlled [80]. In this modern era, bioinformatics approaches play a vital role in the discovery of novel peptides [81]. Traditionally, the peptide design utilizes the homology models or structures along with the docking methods to design the peptides with high affinity against the target proteins [80].

The current findings focused to reveal the potent anti-hypertensive peptides as ACE inhibitors from royal jelly protein (MRJP1) through *in silico* approaches. RMSD and RMSF graphs described the structural stability of MRJP1 in MD simulations at 30 ns along with the quality factor of 92.43%. Peptides were derived by protease hydrolysis, structural characterization /physiological properties, and AHTpin server approaches. The anti-hypertensive potential of scrutinized peptides was measured by QSAR methods of the AHTpin program and the preferable anti-hypertensive candidates with SVM scores ranges from 0.04 to 1.74 were determined. Protein-peptide docking analyses were further carried out to reveal the binding conformations, binding affinities, and potential binding residues. It has also been analyzed that the peptides were embedded with ACE receptor protein and top-ranked 4 peptides were selected

having strong binding affinities *i.e.* 10388 to 9590 docking scores. Top four peptides mainly encompass aromatic amino acid residues including Tyr-Trp-Phe while Tyr was observed as the most abundant amino acid in the selected peptides. Various *in vivo* studies have reported the strong anti-hypertensive activity of aromatic amino acids, particularly Tyr amino acid [82]. It has been suggested that the protease hydrolysis of the royal jelly protein produces many effective ACE inhibitors that would regulate BP.

The molecular docking analyses have the significance of elucidating the interacting residues between the receptor proteins and ligands [83]. Generally, there are three modes of ACE inhibitory peptides as competitive, non-competitive and mixed. The competitive inhibitory peptides possess 2–12 amino acid residues in length and attached at the binding site of ACE. The non-competitive inhibitory peptides showed that the binding other than substrate binding site and affect the ACE enzyme activity. Zn is considered as the significant component of the active site of ACE and ACE activity also depends on Zn [84]. The ACE active site is divided into three binding pockets as $S_1$ (Ala354, Glu384, and Tyr523), S1′ (Glu162) and $S_2$ (Gln281, His353, Lys511, His513, and Tyr520) [85]. The binding stability of peptides at the binding site of the ACE enzyme depends upon hydrogen bonding [86]. Additionally, the involvement of His353, Ala354, Ser355, Glu384, His513, and Pro519 residues are significant for the stability of peptide and enzyme complex while numerous effective peptides have been reported for their interactions at the specified binding sites [87, 88].

The molecular interactional studies of ACE inhibitory peptides are beneficial for the designing and screening of potential novel inhibitory peptides. The reported peptides also present the binding interactions at binding pockets and behave as competitive inhibitory peptides. The top-ranked peptide (EALPHVPIFDR) showed binding interactions in $S_1$ and $S_2$ binding pocket of the ACE enzyme and engaged the significant interacting residues through hydrogen bonding leading to the stability of the complex. The utilized *in silico* approaches provide a novel and potential ACE inhibitors through various distinctive techniques that have the potential to analyze the large-scale conformations through protein-peptide interactions. This leads to an initial step of reducing and eliminating hypertension without drug usage and not to bear their side effects. This could probably be happening only by using those food sources and dietary components, which improves human health and act as preventive measures of these sorts of diseases.

## Conclusions

Contemporary research methods including bioinformatics and proteomic tools applied in current research on peptides from honey protein as a food source and identified the potential anti-hypertensive peptides. It has been demonstrated that the scrutinized peptides EALPHVPIFDR, NLYYSPVASTSLY, PHVPIF, and LYYSPVASTSLYY may have the potential to reduce hypertension with minimal side effects. The reported peptides comprise of aromatic amino acids particularly Tyr and its strong anti-hypertensive activity made the selected peptides a better choice after an extensive *in silico* studies. Even if such food and peptides of proteins are not being able to replace drugs in acute hypertension, they still may have the potential to prevent hypertension.

## Material and methods

### 3.1 Functional information and canonical sequence

The present studies demonstrate the identification of ACE inhibitory peptides from MRJP1 honey protein against hypertension by employing the *in silico* approaches comprising computational 3D modeling, MD simulations, peptides designing, molecular docking analyses, and

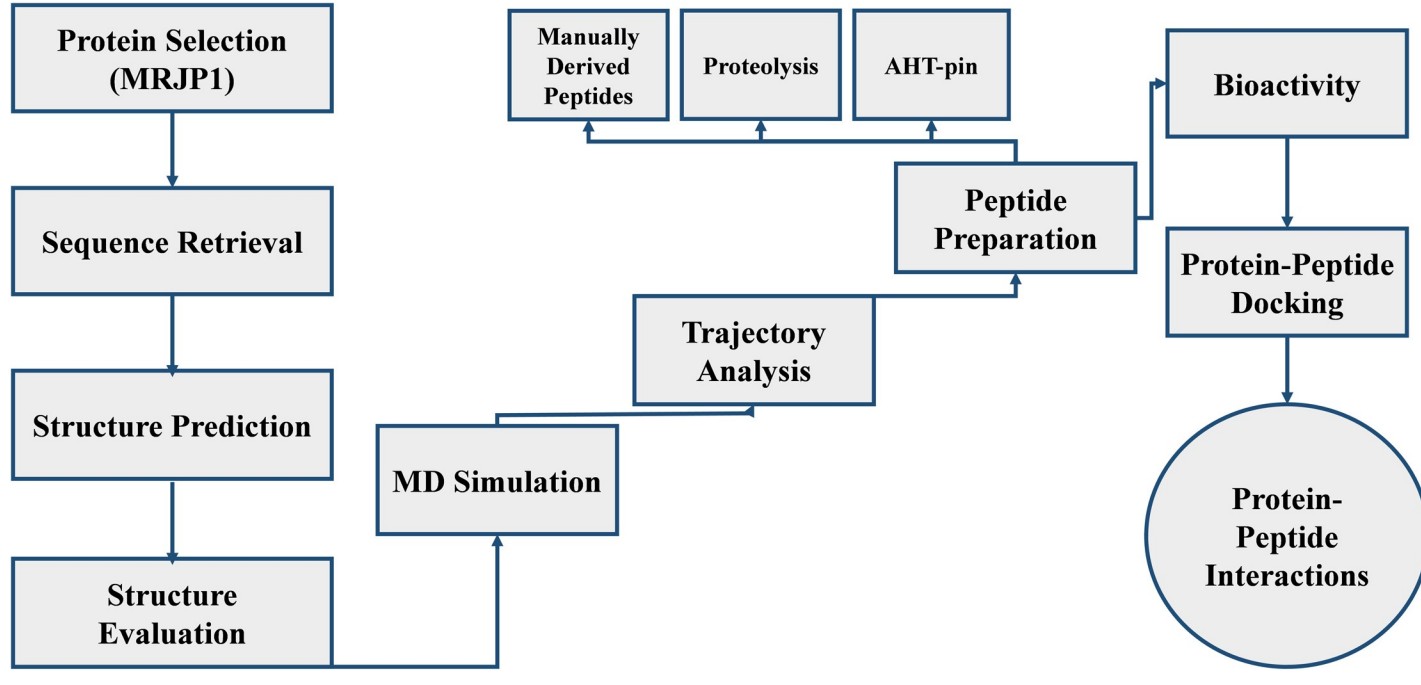

**Fig 6. The methodology of current research work.**

anti-hypertensive activity predictions. The utilized methodology of the current study is presented in a flow chart (Fig 6). The functional information and canonical sequence of MRJP1 in FASTA format were retrieved from UniProt Knowledgebase (http://www.uniprot.org/) having the accession number O18330. The MRJP1 protein sequence was subjected to the protein-protein basic local alignment search tool (BLASTp) [89] against PDB to identify the possible templates. The homology modeling (Modeller 9.14 [54]) and threading based approaches (I-Tasser [50]) were implemented to build the 3D structure of MRJP1. The 3D crystal structure of human ACE was retrieved from PDB (ID: 1O8A) having a resolution of 2 Å determined by the X-ray diffraction method. Errat [90] and Rampage [91] evaluation tools were utilized to evaluate the predicted structure before and after the simulations analyses.

### 3.2 MD simulations

The selected MRJP1 model was subjected to MD simulations by employing AMBER v14 [92] with an ff14SB force field. The simulation analyses were executed in explicit solvent and three-point transferable intermolecular potential (TIP3P) water molecules implemented to solvate the initial structures of a modeled system. Pre-equilibrated elementary cubic box of 78.672 Å $^*$ 84.370 Å $^*$ 79.589 Å was applied to cover the target protein completely that appended 12,397 water molecules. This system amplified the total mass up to 272215.674 amu accompanied by a density of 0.856 g/cm$^3$. The system was neutralized by incorporating the 15 Na$^+$. The comprehensive energy minimization of the solvated protein was carried out for MD simulation experiments. The energy minimization comprising 1500 cycles of conjugate gradient and steepest descent algorithm was executed to eliminate or reduce the energy constraints. SHAKE algorithm was implemented to constrain the hydrogen atoms and bond lengths [93]. A nonbonded cutoff of 10.0 Å with a time step of 0.002 ps was employed by the Berendsen coupling algorithm. Ewald summation method was used to execute the MD simulations for comprehensive electrostatic interactions [94]. The simulation experiments were simulated in initial

equilibration at 1 atm pressure, constant temperature for 300 K and heating time for 500 ps. The simulations for 30 ns were performed and coordinate files were saved after every 5 ns time frame for the structural analyses. PTRAJ module of AMBER generated the output files for the analyses and then visualized by using UCSF Chimera [95]. The obtained results were analyzed by considering various factors including B-Factor, RMSD, and RMSF. The graphs for B-factor, RMSD and RMSF were generated by Microsoft Excel.

## 3.3 Preparation of peptides

The preparation of peptides was performed by three different approaches as structural characterization, proteolysis and AHTpin server [56]. The peptides were manually derived on the basis of anti-hypertensive properties and structural attributes characterizing di-peptides and tri-peptides ACE inhibitors. Di-peptides were composed of amino acids with bulky and hydrophobic side chains, while in tri-peptides, the 1st residue at N-terminal was aromatic, 2nd one was positively charged and the 3rd residue at C-terminal was hydrophobic [25].

**3.3.1 Proteolysis.** Proteolysis was conducted by employing the Peptide Cutter software (http://www.expasy.ch/tools/peptidecutter/) with pepsin and trypsin enzymes individually.

**3.3.2 Peptide derivation.** The anti-hypertensive peptide inhibitors (AHTpin), an online server was used to derive the peptides having anti-hypertensive inhibitory activity by submitting the sequence of MRJP1 to the server. The anti-hypertensive peptides extracted from the above-mentioned techniques were prepared for docking experiments with the receptor protein ACE. Protein-peptide docking analyses were carried out through PatchDock [96] with the parameter of clustering RMSD as 4 to identify the binding affinities of securitized peptides. The top-ranked analyzed complexes were further refined by the Fast Interaction REfinement in the molecular DOCKing (FireDock) server [97] and scrutinized the effective complexes on the basis of their global energy. UCSF Chimera visualization tool was implemented to critically analyze and visualize the peptide interactions and binding pockets accompanied by the bond lengths.

## Supporting information

**S1 Fig. ERRAT quality factor analyses of the predicted structure.**
(TIF)

**S1 File. Table 1: Manually derived peptides with docking scores.** Table 2: Docking Scores for Peptides Generated Using Pepsin Enzyme. Table 3: Docking Scores for Peptides Generated Using Trypsin Enzyme. Table 4: AHTpin peptides docking scores.
(DOCX)

## Author Contributions

**Conceptualization:** Rana Adnan Tahir, Sheikh Arslan Sehgal.

**Data curation:** Rana Adnan Tahir, Yasmine Dali.

**Formal analysis:** Rana Adnan Tahir, Azka Ahmed, Sanaullah Khan, Sheikh Arslan Sehgal.

**Investigation:** Azka Ahmed, Sheikh Arslan Sehgal.

**Methodology:** Rana Adnan Tahir, Muhammad Noaman Yousaf, Azka Ahmed, Yasmine Dali, Sheikh Arslan Sehgal.

**Software:** Afsheen Bashir, Muhammad Noaman Yousaf, Sanaullah Khan, Sheikh Arslan Sehgal.

**Supervision:** Sheikh Arslan Sehgal.

**Validation:** Rana Adnan Tahir, Sheikh Arslan Sehgal.

**Visualization:** Rana Adnan Tahir, Afsheen Bashir, Sanaullah Khan, Sheikh Arslan Sehgal.

**Writing – original draft:** Sheikh Arslan Sehgal.

**Writing – review & editing:** Rana Adnan Tahir, Afsheen Bashir, Muhammad Noaman Yousaf, Sanaullah Khan, Sheikh Arslan Sehgal.

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
