## [Decision Letter · Decision Letter 0]

8 Aug 2019

PONE-D-19-16606

In silico Identification of Angiotensin Converting Enzyme inhibitory Peptides from MRJP1

PLOS ONE

Dear Dr Sehgal,

Thank you for submitting your manuscript to PLOS ONE. After careful consideration, we feel that it has merit but does not fully meet PLOS ONE’s publication criteria as it currently stands. Therefore, we invite you to submit a revised version of the manuscript that addresses the points raised during the review process.

We would appreciate receiving your revised manuscript by 30th September. To enhance the reproducibility of your results, we recommend that if applicable you deposit your laboratory protocols in protocols.io, where a protocol can be assigned its own identifier (DOI) such that it can be cited independently in the future. For instructions see: http://journals.plos.org/plosone/s/submission-guidelines#loc-laboratory-protocols

We look forward to receiving your revised manuscript.

Kind regards,

Ghulam Md Ashraf, Ph.D.

Academic Editor

PLOS ONE

Journal Requirements:

1. Thank you for including your funding statement; "No"

Please provide an amended Funding Statement that declares *all* the funding or sources of support received during this specific study (whether external or internal to your organization) as detailed online in our guide for authors at http://journals.plos.org/plosone/s/submit-now.  

Please state what role the funders took in the study.  If any authors received a salary from any of your funders, please state which authors and which funder. If the funders had no role, please state: "The funders had no role in study design, data collection and analysis, decision to publish, or preparation of the manuscript."

2. Thank you for including your competing interests statement; "No"

Additional Editor Comments (if provided):

The authors are advised to address all the concerns raised by the reviewers.

Reviewers' comments:

Reviewer's Responses to Questions

**Comments to the Author**

1. Is the manuscript technically sound, and do the data support the conclusions?

Reviewer #1: Partly

Reviewer #2: Partly

2. Has the statistical analysis been performed appropriately and rigorously? 

Reviewer #1: Yes

Reviewer #2: Yes

3. Have the authors made all data underlying the findings in their manuscript fully available?

Reviewer #1: Yes

Reviewer #2: Yes

4. Is the manuscript presented in an intelligible fashion and written in standard English?

Reviewer #1: Yes

Reviewer #2: Yes

5. Review Comments to the Author

Reviewer #1: The manuscript submitted by Rana Tahir et al depicted "In silico Identification of Angiotensin Converting Enzyme inhibitory Peptides from MRJP1", established a systematic computational workflow to screen ACE inhibitor by Homology modeling, Molecular dynamic and docking, and predicted peptide “EALPHVPIFDR” has higher anti-hypertensive activity, and we looking forward to some deeper and new-finding based on that. This work is innovative and interesting. And the methods are also standard. In conclusion, this paper can be accepted, but there are still some places need to be revised:

1:Is the time of 30ns too short for protein structure stabilization and optimization using molecular dynamics?

2. It is suggested to incorporate the flow chart of the applied methodology for better understanding.

3: Authors stated that "The values of RMSD and RMSF denoted that the MRJP1 has rare loop structures at corresponding residues", What is the basis of this argument? References need to be updated.

4. The label of Figure 5 is difficult to see particularly yellow color. Change it to visible labeling and yellow color to any other one

5. Authors stated that "Mainly, PubChem, CDK-fingerprint, XLogP, electrotopological state atom type and auto-correlation descriptors were implemented..." and "while tri- peptide QSAR model was developed primarily by KlekotaRoth fingerprint count, PubChem fingerprint, CDK graph only fingerprint and extended fingerprint descriptors....". What is the basis of selecting these different descriptors for di-peptide or tri-peptide? References need to be updated. It is recommended to add a detailed description for modelling process of SVM, including feature engineeing (how to select or exclude features) and relevant diagrams.

6. Auther stated that "The peptide “EALPHVPIFDR” showed strong binding affinity and higher anti-hypertensive activity along with the global energy of -58.29 and docking score of 9590". The authors are off to a good start, however, this study requires additional experiments, such as implementing MD on complex structure from docking and then computing

binding free energy by MMPBSA. It would be better if auther supplement the simple biological test in vitro.

Reviewer #2: The authors present a purely computational study to identify Novel Angiotensin‐Converting Enzyme Inhibitory Peptides Derived from honey protein MRJP1. They tried to make a reliable structural model of MRJP1. Later, the top ranked derived peptides was docked against MRJP1 to analyse the molecular interactions. The fact that several modeling services have been employed, as well as their combination with MD refinement denote considerable care. However, the underlying problem remains the lack of independent confirmation of the docking poses, which is possibly the weakest link.

Major issues:

1) My main problem with this manuscript is that I do not understand who might take advantage of the reported findings: i) Biochemists/Molecular biologists might be tempted to validate In vitro the top ranked peptides if they have the system in place already, and peptides need to be synthesized and to make it proteolytic resistant which is the major challenge in peptide designing. ii) Computational fellows might want the presented pipeline to be validated before applying the method to other systems and iii) Chemists as well will not start any optimization before an experimental evidence.

2) The other major concern relates with their homology modeling approach. When they had an x-ray resolved crystal structure of Major Royal Jelly Protein 1 Oligomer readily available (PDB ID: 5YYL against their mentioned uniprot sequence ID: O18330) then there was absolutely no need to perform homology modeling, MD refinement and later model validation analysis. They should revise the study by taking into account the crystal structure and perform further analysis. Later, they should perform MD simulation of MRJP1 with/without bound peptide to examine the considerable influence of bound peptide. They should also perform the actual MM-GBSA calculations to better explore the binding free energy calculations in the presence of explicit solvent. The only docking conformations are not enough to estimate the most plausible interactions.

3) Another comment on their work relates with a suggestion if one considers a homology model. On what basis a 30ns timescale was selected? A careful reflection on RMSD revealed a gradual expansion even after 30 ns. In general, especially to estimate the protein’s stability, a simulation must be long enough to converge the dynamics of interest and exhibit equilibrium sampling. Everything depends on what you're trying to study, and depends again on the size of the system. For example, two small proteins with only one domain could perhaps be studied in a matter of tens or hundreds of nanoseconds. Complexes of multi-domain proteins may require microseconds or milliseconds depending on the time scale of domain motion. If there was a need of homology modeling (provided the template with >70% identity), they should have examined a backbone stability unless RMSD show convergence up to 10 ns at least. Later, they should perform clustering analysis and take the most representative conformation (with RMSD < 1Å) from the largest cluster to declare it a reasonable model. To better understand the timescale, author should consider this article:

Zwier, M.C. and Chong, L.T., 2010. Reaching biological timescales with all-atom molecular dynamics simulations. Current opinion in pharmacology, 10(6), pp.745-752.

4) I really enjoyed reading their discussion. After considering all above points in their revised study, the authors should compare their findings with the previously reported ACE inhibitory peptides. Quite a few articles are already published which have the same methodology as authors presented in their study. For example:

Yu, Z., Fan, Y., Zhao, W., Ding, L., Li, J. and Liu, J., 2018. Novel Angiotensin‐Converting Enzyme Inhibitory Peptides Derived from Oncorhynchus mykiss Nebulin: Virtual Screening and In Silico Molecular Docking Study. Journal of food science, 83(9), pp.2375-2383.

Yu, Z., Chen, Y., Zhao, W., Li, J., Liu, J. and Chen, F., 2018. Identification and molecular docking study of novel angiotensin‐converting enzyme inhibitory peptides from Salmo salar using in silico methods. Journal of the science of food and agriculture, 98(10), pp.3907-3914.

Vukic, V.R., Vukic, D.V., Milanovic, S.D., Ilicic, M.D., Kanuric, K.G. and Johnson, M.S., 2017. In silico identification of milk antihypertensive di-and tripeptides involved in angiotensin I–converting enzyme inhibitory activity. Nutrition research, 46, pp.22-30.

Others include:

Yu, Z., Wu, S., Zhao, W., Ding, L., Shiuan, D., Chen, F., Li, J. and Liu, J., 2018. Identification and the molecular mechanism of a novel myosin-derived ACE inhibitory peptide. Food & function, 9(1), pp.364-370.

Wang, C., Tu, M., Wu, D., Chen, H., Chen, C., Wang, Z. and Jiang, L., 2018. Identification of an ACE-Inhibitory Peptide from Walnut Protein and Its Evaluation of the Inhibitory Mechanism. International journal of molecular sciences, 19(4), p.1156.

Tu, M., Wang, C., Chen, C., Zhang, R., Liu, H., Lu, W., Jiang, L. and Du, M., 2018. Identification of a novel ACE-inhibitory peptide from casein and evaluation of the inhibitory mechanisms. Food chemistry, 256, pp.98-104.

Yu, Z., Fan, Y., Zhao, W., Ding, L., Li, J. and Liu, J., 2018. Novel Angiotensin‐Converting Enzyme Inhibitory Peptides Derived from Oncorhynchus mykiss Nebulin: Virtual Screening and In Silico Molecular Docking Study. Journal of food science, 83(9), pp.2375-2383.

By providing a comparison with the previously reported ACE inhibitory peptides, a fruitful discussion can be presented.

6. PLOS authors have the option to publish the peer review history of their article (what does this mean?). If published, this will include your full peer review and any attached files.

Reviewer #1: No

Reviewer #2: Yes: Muhammad Usman Mirza

---

## [Author Response · Author response to Decision Letter 0]

3 Oct 2019

We are very thankful to Editor and Reviewers for valuable comments to improve the quality of the manuscript. Authors critically studied the comments by reviewers and solved the comments and improved the manuscript. 

Reviewer’s Comments

Reviewer 1: 

Comment 1: Is the time of 30 ns too short for protein structure stabilization and optimization using molecular dynamics?

Answer: The major fluctuations were observed only at the initial phases of MD simulations, afterward, structure became stabilized till 30 ns MD simulations. RMSD graph in manuscript presenting the fluctuations and stability of 3D model and there were no such variations in a model at the last frames of simulations and therefore optimized model was obtained at 30 ns. 

Comment 2: It is suggested to incorporate the flow chart of the applied methodology for better understanding.

Answer: As per suggestion, flow chat of applied methodology has been incorporated at the appropriate position. 

Comment 3: Authors stated that "The values of RMSD and RMSF denoted that the MRJP1 has rare loop structures at corresponding residues", What is the basis of this argument? References need to be updated.

Answer: Thanks for your highlighting, as per suggestion the sentence has been updated to remove the ambiguity and for better understanding.

Comment 4: The label of Figure 5 is difficult to see particularly yellow color. Change it to visible labeling and yellow color to any other one. 

Answer: As per suggestion, the yellow color has been replaced with another color for more visibility. 

Comment 5: Authors stated that "Mainly, PubChem, CDK-fingerprint, XLogP, electrotopological state atom type and auto-correlation descriptors were implemented..." and "while tri- peptide QSAR model was developed primarily by KlekotaRoth fingerprint count, PubChem fingerprint, CDK graph only fingerprint and extended fingerprint descriptors....". What is the basis of selecting these different descriptors for di-peptide or tri-peptide? References need to be updated. It is recommended to add a detailed description for modelling process of SVM, including feature engineeing (how to select or exclude features) and relevant diagrams.

Answer: The utilized methodology of current research has been incorporated including the peptides derivations and evaluation. AHTpin is an in silico method to predict and design the antihypertensive peptides that is utilized in the current study to derive the antihypertensive peptides followed by the activities (SVM Scores) of already derived peptides through other methods. Here, the descriptors utilized to develop an algorithm of AHTpin tool and QSAR models are described. QSAR models are statistical tools built to correlate the biological bioactivity with descriptors of compounds. The already available QSAR models were utilized in current study and algorithms or working of these methods are discussed at appropriate positions in the manuscript. The references of current method have been updated as per suggestion. 

Comment 6: Author stated that "The peptide “EALPHVPIFDR” showed strong binding affinity and higher anti-hypertensive activity along with the global energy of -58.29 and docking score of 9590". The authors are off to a good start, however, this study requires additional experiments, such as implementing MD on complex structure from docking and then computing

binding free energy by MMPBSA. It would be better if auther supplement the simple biological test in vitro.

Answer: The detailed in silico analyses were performed to determine the potential antihypertensive peptides from royal jelly protein verified through docking analysis and regression models (QSAR models) used in AHTpin program. SVM scores are predicted by utilizing the models built on in vitro bioactivities of peptides followed by different docking tools. First, PatchDock identifies the docking transformations and afterwards evaluate each transformation through scoring function based on geometrical fit and atomic solvation energy. The top docking solutions are further refined and optimized by using the FireDock server which delivers the flexible refinement and scoring of docking solutions. It includes optimization of side-chain conformations and rigid-body orientation and allows a high-throughput refinement. Our lab mainly works on computational drug designing and has fewer facilities for wet lab analyses.

Reviewer 2: 

The authors present a purely computational study to identify Novel Angiotensin‐Converting Enzyme Inhibitory Peptides Derived from honey protein MRJP1. They tried to make a reliable structural model of MRJP1. Later, the top ranked derived peptides was docked against MRJP1 to analyse the molecular interactions. The fact that several modeling services have been employed, as well as their combination with MD refinement denote considerable care. However, the underlying problem remains the lack of independent confirmation of the docking poses, which is possibly the weakest link.

Comment 1: My main problem with this manuscript is that I do not understand who might take advantage of the reported findings: i) Biochemists/Molecular biologists might be tempted to validate In vitro the top ranked peptides if they have the system in place already, and peptides need to be synthesized and to make it proteolytic resistant which is the major challenge in peptide designing. ii) Computational fellows might want the presented pipeline to be validated before applying the method to other systems and iii) Chemists as well will not start any optimization before an experimental evidence.

Answer: Biochemists, molecular biologists, computational biologists, chemists and other researchers take advantage from the current findings as in detailed in silico analyses as presented in this manuscript has 50-60% chances of success. Every researcher has to perform his task as biochemists are unable to do this extensive in silico study as computational biologists are unable to do in vitro analyses. It is better for the researchers to synthesize and validates the findings through wet lab instead of trying millions of peptides with huge budget and time. The computational biologists who have good expertise in their domain will trust the analyses after reading the detailed findings and researchers have to trust each other work and efforts. Computational biologists will also learn from the utilized methodology in other relevant projects. There is not a single research that has no benefit for others and the researchers always find the benefits from the other researcher’s findings.

Comment 2: Another comment on their work relates with a suggestion if one considers a homology model. On what basis a 30 ns timescale was selected? A careful reflection on RMSD revealed a gradual expansion even after 30 ns. In general, especially to estimate the protein’s stability, a simulation must be long enough to converge the dynamics of interest and exhibit equilibrium sampling. Everything depends on what you're trying to study, and depends again on the size of the system. For example, two small proteins with only one domain could perhaps be studied in a matter of tens or hundreds of nanoseconds. Complexes of multi-domain proteins may require microseconds or milliseconds depending on the time scale of domain motion. If there was a need of homology modeling (provided the template with >70% identity), they should have examined a backbone stability unless RMSD show convergence up to 10 ns at least. Later, they should perform clustering analysis and take the most representative conformation (with RMSD < 1Å) from the largest cluster to declare it a reasonable model. To better understand the timescale, author should consider this article.

Answer: Refer to the comment number 1 of reviewer 1. The mentioned articles have been added to the manuscript at appropriate positions as per suggestion.

Comment 3: I really enjoyed reading their discussion. After considering all above points in their revised study, the authors should compare their findings with the previously reported ACE inhibitory peptides. Quite a few articles are already published which have the same methodology as authors presented in their study.

Answer: Thanks for the appreciation the Discussion section. All the suggested articles are added to the manuscript at appropriate positions as per suggestion. 

*Changes are mentioned in colors in a manuscript.

---

## [Decision Letter · Decision Letter 1]

24 Oct 2019

PONE-D-19-16606R1

In silico Identification of Angiotensin Converting Enzyme inhibitory Peptides from MRJP1

PLOS ONE

Dear Dr. Sehgal,

Thank you for submitting your manuscript to PLOS ONE. After careful consideration, we feel that it has merit but does not fully meet PLOS ONE’s publication criteria as it currently stands. Therefore, we invite you to submit a revised version of the manuscript that addresses the points raised during the review process.

We would appreciate receiving your revised manuscript by 25th November 2019. To enhance the reproducibility of your results, we recommend that if applicable you deposit your laboratory protocols in protocols.io, where a protocol can be assigned its own identifier (DOI) such that it can be cited independently in the future. For instructions see: http://journals.plos.org/plosone/s/submission-guidelines#loc-laboratory-protocols

We look forward to receiving your revised manuscript.

Kind regards,

Ghulam Md Ashraf, Ph.D.

Academic Editor

PLOS ONE

Reviewers' comments:

Reviewer's Responses to Questions

**Comments to the Author**

1. If the authors have adequately addressed your comments raised in a previous round of review and you feel that this manuscript is now acceptable for publication, you may indicate that here to bypass the “Comments to the Author” section, enter your conflict of interest statement in the “Confidential to Editor” section, and submit your "Accept" recommendation.

Reviewer #1: All comments have been addressed

Reviewer #2: (No Response)

2. Is the manuscript technically sound, and do the data support the conclusions?

Reviewer #1: Yes

Reviewer #2: Partly

3. Has the statistical analysis been performed appropriately and rigorously? 

Reviewer #1: Yes

Reviewer #2: Yes

4. Have the authors made all data underlying the findings in their manuscript fully available?

Reviewer #1: Yes

Reviewer #2: Yes

5. Is the manuscript presented in an intelligible fashion and written in standard English?

Reviewer #1: Yes

Reviewer #2: Yes

6. Review Comments to the Author

Reviewer #1: The manuscript submitted by Rana Tahir et al depicted "In silico Identification of Angiotensin Converting Enzyme inhibitory Peptides from MRJP1", established a systematic computational workflow to screen ACE inhibitor by Homology modeling, Molecular dynamic and docking, and predicted peptide “EALPHVPIFDR” has higher anti-hypertensive activity, and we looking forward to some deeper and new-finding based on that. This work is innovative and interesting. And the methods are also standard. The response is basically reasonable. In conclusion, this paper can be accepted.

Reviewer #2: The authors improved the manuscript but I think authors simply overlooked one major concern as follows (2nd comment in my first revision):

1. "2). The other major concern relates with their homology modeling approach. When they had an x-ray resolved crystal structure of Major Royal Jelly Protein 1 Oligomer readily available (PDB ID: 5YYL against their mentioned uniprot sequence ID: O18330) then there was absolutely no need to perform homology modeling, MD refinement and later model validation analysis. They should revise the study by taking into account the crystal structure and perform further analysis. Later, they should perform MD simulation of MRJP1 with/without bound peptide to examine the considerable influence of bound peptide. They should also perform the actual MM-GBSA calculations to better explore the binding free energy calculations in the presence of explicit solvent. The only docking conformations are not enough to estimate the most plausible interactions"

Once MD simulations has performed then its easy to calculate the MM-GBSA/MM-PBSA. Authors can refer a tutorial from the link below as they used AMBER 14 simulation package:

https://ambermd.org/tutorials/advanced/tutorial3/py_script/section2.htm

2. The advantage of having a crystal structure in this study (as it is readily available) will significantly improved MD simulation analysis in terms of overall stability with bound peptide and binding free energy calculations.

The manuscript should publish after the incorporation of above mentioned suggestions/comments.

7. PLOS authors have the option to publish the peer review history of their article (what does this mean?). If published, this will include your full peer review and any attached files.

Reviewer #1: No

Reviewer #2: Yes: Muhammad Usman Mirza

---

## [Author Response · Author response to Decision Letter 1]

12 Dec 2019

Response to Reviewer’s

In silico Identification of Angiotensin-Converting Enzyme inhibitory Peptides from MRJP1

PLOS ONE: PONE-D-19-16606

We are very thankful to the Editor and Reviewers for valuable comments to improve the quality of the manuscript. Authors critically studied the comments by reviewers and solved the comments and improved the manuscript. 

Reviewer’s Comments

Reviewer 1: The manuscript submitted by Rana Tahir et al depicted "In silico Identification of Angiotensin Converting Enzyme Inhibitory Peptides from MRJP1", established a systematic computational workflow to screen ACE inhibitor by Homology modeling, Molecular dynamic and docking, and predicted peptide “EALPHVPIFDR” has higher anti-hypertensive activity, and we looking forward to some deeper and new-finding based on that. This work is innovative and interesting. And the methods are also standard. The response is basically reasonable. In conclusion, this paper can be accepted.

Answer: Thanks to the reviewer for accepting the manuscript. 

Reviewer 2: 

Comment 1: 

The other major concern relates with their homology modeling approach. When they had an x-ray resolved crystal structure of Major Royal Jelly Protein 1 Oligomer readily available (PDB ID: 5YYL against their mentioned uniprot sequence ID: O18330) then there was absolutely no need to perform homology modeling, MD refinement and later model validation analysis. They should revise the study by taking into account the crystal structure and perform further analysis. Later, they should perform MD simulation of MRJP1 with/without bound peptide to examine the considerable influence of bound peptide. They should also perform the actual MM-GBSA calculations to better explore the binding free energy calculations in the presence of an explicit solvent. The only docking conformations are not enough to estimate the most plausible interactions". Once MD simulations has performed then its easy to calculate the MM-GBSA/MM-PBSA. 

Answer: It is a good suggestion to use the crystal structure for the analyses. We predicted and simulated the model to derive the peptides manually and also through pepfold tool but the whole predicted structure are not utilized in mainstream analyses. Multiple techniques were employed to extract the potential di and tri-peptides from the MRJP1 model. Furthermore, the predicted structure was then used to cross verify the model and peptides (as the x-ray crystallographic structure was online after) and 95% results were similar. As already mentioned in the previous revision, the reported peptides were docked and refined for the verification of analyses. 

The extracted peptides were docked and simulated through PatchDock and Firedock and the global free energy was calculated to identify the potential peptides. The hundred poses of each complex were selected from the PatchDock based on scoring function that is combination of pairwise shape complementarity, desolvation and electrostatic energy. The refinement of selected poses was performed by FireDock that refines and scores them according to an energy function, spending about 3.5 seconds per candidate solution. The rearrangement of the side-chains, the relative position of docking partners is refined by Monte Carlo minimization of the binding score function and the refined candidates are ranked by the binding score. FireDock score includes Atomic Contact Energy, softened van der Waals interactions, partial electrostatics and additional estimations of the binding free energy. The final selected protein-peptide complexes are identified through extensive modeling, docking and simulations strategies and it has been reported that these docking and refinement, algorithms based on multiple energy functions are considered as effective approaches to identify the potential docking solutions. Global free energy and MM-GBSA/MM-PBSA have similar algorithms to calculate the free energy after docking and simulation. The suggested experiment was already performed through software instead of AMBER and reliable results were observed.

Comment 2: 

2. The advantage of having a crystal structure in this study (as it is readily available) will significantly improved MD simulation analysis in terms of overall stability with bound peptide and binding free energy calculations. 

Answer: Thanks for the suggestion as the same analyses were already performed in the analyses and explain briefly in the manuscript. 

Hope, these modifications are sufficient to resolve the queries, If Editor or reviewer still feel to incorporate the interactions of peptides extracted from the experimental model in the main manuscript, we can update that results. 

*Changes are mentioned in colors in a manuscript.

---

## [Decision Letter · Decision Letter 2]

13 Jan 2020

In silico Identification of Angiotensin Converting Enzyme inhibitory Peptides from MRJP1

PONE-D-19-16606R2

Dear Dr. Sehgal,

We are pleased to inform you that your manuscript has been judged scientifically suitable for publication and will be formally accepted for publication once it complies with all outstanding technical requirements.

With kind regards,

Ghulam Md Ashraf, Ph.D.

Academic Editor

PLOS ONE

Additional Editor Comments (optional):

Reviewers' comments:

Reviewer's Responses to Questions

**Comments to the Author**

1. If the authors have adequately addressed your comments raised in a previous round of review and you feel that this manuscript is now acceptable for publication, you may indicate that here to bypass the “Comments to the Author” section, enter your conflict of interest statement in the “Confidential to Editor” section, and submit your "Accept" recommendation.

Reviewer #2: All comments have been addressed

2. Is the manuscript technically sound, and do the data support the conclusions?

Reviewer #2: Partly

3. Has the statistical analysis been performed appropriately and rigorously? 

Reviewer #2: Yes

4. Have the authors made all data underlying the findings in their manuscript fully available?

Reviewer #2: Yes

5. Is the manuscript presented in an intelligible fashion and written in standard English?

Reviewer #2: Yes

6. Review Comments to the Author

Reviewer #2: Although the manuscript lack the important binding free energy calculations after MD simulations but the presented data is still publishable. In future studies, the authors should include MM-GBSA/PBSA calculations as an integral part after simple more unpredictable docking studies.

7. PLOS authors have the option to publish the peer review history of their article (what does this mean?). If published, this will include your full peer review and any attached files.

Reviewer #2: No

---

## [Editor Report · Acceptance letter]

17 Jan 2020

PONE-D-19-16606R2 

*In silico* Identification of Angiotensin-Converting Enzyme inhibitory Peptides from MRJP1 

Dear Dr. Sehgal:

I am pleased to inform you that your manuscript has been deemed suitable for publication in PLOS ONE. Congratulations! Your manuscript is now with our production department. 

With kind regards,

on behalf of

Dr. Ghulam Md Ashraf 

Academic Editor

PLOS ONE